# NEURAL IMAGE-BASED AVATARS: GENERALIZABLE RADIANCE FIELDS FOR HUMAN AVATAR MODELING

**Youngjoong Kwon**[1], **Dahun Kim**[2], **Duygu Ceylan**[3], **Henry Fuchs**[1]
[1]University of North Carolina at Chapel Hill. [2]Google Research, Brain Team. [3]Adobe Research.
{youngjoong,fuchs}@cs.unc.edu  {mcahny}@google.com  {ceylan}@adobe.com

## ABSTRACT

We present a method that enables synthesizing novel views and novel poses of arbitrary human performers from sparse multi-view images. A key ingredient of our method is a hybrid appearance blending module that combines the advantages of the implicit body NeRF representation and image-based rendering. Existing generalizable human NeRF methods that are conditioned on the body model have shown robustness against the geometric variation of arbitrary human performers. Yet they often exhibit blurry results when generalized onto unseen identities. Meanwhile, image-based rendering shows high-quality results when sufficient observations are available, whereas it suffers artifacts in sparse-view settings. We propose Neural Image-based Avatars (NIA) that exploits the best of those two methods: to maintain robustness under new articulations and self-occlusions while directly leveraging the available (sparse) source view colors to preserve appearance details of new subject identities. Our hybrid design outperforms recent methods on both in-domain identity generalization as well as challenging cross-dataset generalization settings. Also, in terms of the pose generalization, our method outperforms even the per-subject optimized animatable NeRF methods.

## 1 INTRODUCTION

Acquisition of 3D renderable full-body avatars is critical for applications to virtual reality, telepresence and human modeling. While early solutions have required heavy hardware setups such as dense camera rigs or depth sensors, recent neural rendering techniques have achieved significant progress to a more scalable and low-cost solution. Notably, neural radiance fields (NeRF) based methods facilitated by the parametric body prior Loper et al. (2015) require only sparse camera views to enable visually pleasing free-view synthesis Peng et al. (2021b); Kwon et al. (2021); Zhao et al. (2022); Cheng et al. (2022) or pose animation Peng et al. (2021a); Su et al. (2021) of the human avatar.

Still, creating a full-body avatar from sparse images (*e.g.*, three snaps) of a person is a challenging problem due to the complexity and diversity of possible human appearances and poses. Most existing methods Peng et al. (2021b;a) are therefore focusing on person-specific setting which requires a dedicated model optimization for each new subject it encounters. More recent methods explore generalizable human NeRF representations Peng et al. (2021b); Raj et al. (2021b); Kwon et al. (2021); Zhao et al. (2022) by using pixel-aligned features in a data-driven manner. Among them, Kwon et al. (2021) and Chen et al. (2022) specifically exploit the *body surface feature* conditioned NeRF (*i.e.*, pixel-aligned features anchored at the SMPL vertices) which helps robustness to various articulations while obviating the need for the 3D supervision Zhao et al. (2022); Cheng et al. (2022). Nevertheless, these body surface feature conditioned NeRFs still suffer blur artifacts when generalizing onto unseen subject identities with complex poses. Also, an extra effort on video-level feature aggregation is required in Kwon et al. (2021) to compensate for the sparsity of input views.

In this paper, we propose Neural Image-based Avatars (NIA) that generalizes novel view synthesis and pose animation for arbitrary human performers from a sparse set of still images. It is a hybrid framework that combines body surface feature conditioned NeRF (*e.g.*, Kwon et al. (2021)) and image-based rendering techniques (*e.g.*, Wang et al. (2021)). While the former helps in robust representation of different body shapes and poses, the image-based rendering helps preserving the color and texture details from the source images. This can complement the NeRF predicted colors

which are often blur and inaccurate in generalization settings (cross-identity as well as cross-dataset generalization) as shown in figure 2 and figure 5. To leverage the best of both worlds, we propose a neural appearance blending scheme that learns to adaptively blend the NeRF predicted colors with the direct source image colors. Last but not least, by deforming the learned NIA representation based on the skeleton-driven transformations Lewis et al. (2000); Kavan et al. (2007), we enable plausible pose animation of the learned avatar.

To demonstrate the efficacy of our NIA method, we experiment on ZJU-MoCap Peng et al. (2021b) and MonoCap Habermann et al. (2020; 2021) datasets. First, experiments show that our method outperforms the state-of-the-art Neural Human Performer Kwon et al. (2021) and GP-NeRF Chen et al. (2022) in novel view synthesis task. Furthermore, we study the more challenging cross-dataset generalization by evaluating the zero-shot performance on the MonoCap Habermann et al. (2020; 2021) datasets, where we clearly outperform the previous methods. Finally, we evaluate on the pose animation task, where our NIA tested on *unseen subjects* achieves better pose generalization than the *per-subject* trained A-NeRF Su et al. (2021) and Animatable-NeRF Peng et al. (2021a) that are tested on the *seen* training subjects. The ablation studies demonstrate that the proposed modules of our NIA collectively contribute to the high-quality rendering for arbitrary human subjects.

## 2 RELATED WORK

Combined with Neural Radiance Fields (NeRF) Mildenhall et al. (2020), human reconstruction research has shown unprecedented development Pumarola et al. (2020); Park et al. (2021a;b). Human priors are utilized to enable robust reconstruction of face and body Gao et al. (2020); Gafni et al. (2021); Peng et al. (2021b). However, these methods are per-subject optimized, and cannot model the motions that are not seen during training. Therefore, subsequent works have been focusing on generalization in two directions: pose and subject identity.

**Pose generalization.** Su et al. (2021) utilize a joint-relative encoding for dynamic articulations. Noguchi et al. (2021) explicitly associate 3D points to body parts. Chen et al. (2021) and Peng et al. (2021a) deform the target pose space queries into the canonical space to obtain the color and density values. Liu et al. (2021) leverages normal map as the dense pose cue. Xu et al. (2021) learns deformable signed distance field. Weng et al. (2022) and Peng et al. (2022) decompose the human deformation into articulation-driven and non-rigid deformations. Su et al. (2022) and Zheng et al. (2022) utilize the joint-specific local radiance fields. Despite the significant progress in pose generalization, they still focus on a subject-specific setting which requires training a single model for each subject. In this paper, we tackle generalization across both poses and subject identities.

**Subject identity generalization.** The use of image-conditioned or pixel-aligned features Yu et al. (2020); Wang et al. (2021) has allowed generalized neural human representations from sparse camera views. Raj et al. (2021b) use camera-encoded pixel-aligned features for face view synthesis. Kwon et al. (2021) aggregate temporal features by anchoring them onto the SMPL body vertice to complement the sparse input views. Chen et al. (2022) also leverages body surface features to enable full-body synthesis. Zhao et al. (2022) and Cheng et al. (2022) also propose the neural blending of NeRF prediction with source view colors. Specifically, they use the pixel-aligned features-conditioned NeRF as their implicit body representation. However, pixel-aligned features alone are prone to errors under complex poses as reported in Kwon et al. (2021). Therefore, their method require 3D supervision (*e.g.*, depth, visibility) or per-subject finetuning. In contrast to these methods, we aim at generalizing human modeling without relying on the 3D supervision or per-subject finetuning, but by using only RGB supervision.

**Other (non NeRF-based) methods.** Deferred neural rendering based methods Thies et al. (2019); Raj et al. (2021a); Grigorev et al. (2021) enable fast synthesis of person-specific avatar by combining the traditional graphics pipeline with the neural texture maps. Grigorev *et al.* further finetunes to deal with unseen subjects. In contrast, we focus on the generalization without any finetuning. Saito et al. (2019; 2020); He et al. (2021) leverage pixel-aligned features to enable 3d reconstruction from a single image. However, they require 3D groundtruth supervision. Habermann et al. (2019); Liu et al. (2020); Habermann et al. (2021); Bagautdinov et al. (2021) generate high-quality non-rigid deformation, but they require the template mesh optimization. Aliev et al. (2019); Wu et al. (2020) anchor features on the point clouds and render them with differentiable rasterizer. However, they

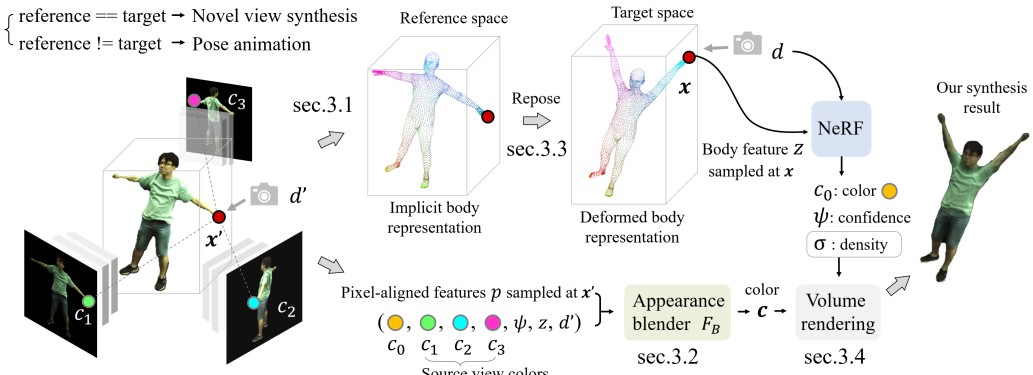

Figure 1: **Overview of Neural Image-based Avatars (NIA) method.** Given a sparse view images of an unseen person in the reference space, NIA instantly creates a human avatar for novel view synthesis and pose animation. NIA is a hybrid approach combining the SMPL-based implicit body representation and the image-based rendering method. Our appearance blender learns to adaptively blend the predictions from the two components. Note that if the reference space and the target pose spaces are identical (*i.e.*, no pose difference, $x = x'$, $d = d'$), the task is novel view synthesis. Otherwise, it is a pose animation task where we deform the NIA representation to repose the learned avatar.

require sufficient amount of point clouds that cannot be obtained from our sparse input view setting. In this work, we mainly focus on and compare with the NeRF-based methods.

## 3 METHOD

The overview of our NIA is given in figure 1. Our goal is to obtain a Neural Image-based Avatars (NIA) which after training, can directly synthesize novel views and novel poses of an unseen person in a feed-forward manner from very sparse input views. To deal with complex and diverse human appearances and poses, we propose a *hybrid* framework that leverages the advantages of two representations. The first is a neural radiance fields based body representation conditioned on a parametric human model (SMPL Loper et al. (2015)), which can provide the body geometry and pose-dependent deformations (section 3.1). Inspired by Wang et al. (2021), the second is obtained by neural image-based rendering which can improve the fine-details of the result by directly fetching the pixel colors from the source view images. A core of our design is a neural appearance blending scheme that learns to blend predictions from the two representations and outputs the final synthesis results (section 3.2). Given a set of sparse view images of an arbitrary person, our learned NIA not only performs novel view synthesis, but also reposing by deforming the representation of the avatar at capture time into the desired target pose (section 3.3).

We note that we assume the calibration parameters of the multi-view input images and the foreground human region masks are known, and the fitted SMPL models are available.

### 3.1 BASELINE: GENERALIZABLE IMPLICIT BODY REPRESENTATION

Inspired by NHP Kwon et al. (2021) and GP-NeRF Chen et al. (2022), our implicit body NeRF representation follows the design of body surface feature conditioned NeRF (*i.e.*, pixel-aligned features anchored at SMPL vertices.

Given $N$ available source views, we extract pixel-aligned image features Saito et al. (2019) from each view $n$ and attach them to the SMPL vertices, to construct a per-view coarse feature set $P_n$. Each vertex location is projected to the image feature plane and the corresponding pixel-aligned feature is computed by bilinearly interpolating the grid pixel values. Since $P_n$ is sparse in 3D to represent the whole human body, we use 3D sparse convolutions Liu et al. (2015) to diffuse the features into a denser feature volume. The sparse convolutions learn to extrapolate the sparse feature points $P_n$ to build a dense body feature volume $\tilde{P}_n$ that better adapts to the target body shape. For each query point $\mathbf{x}$, we sample the corresponding body feature from $\tilde{P}_n$. The sampled features from $N$ source

views are then processed by a view-wise cross-attention with the pixel-aligned features to obtain the multi-view aware body feature $\mathbf{z}_n$. Besides the final density $\sigma$ and color value $\mathbf{c_0}$, we also propose to predict the confidence $\psi \in [0, 1]$ of its color prediction as:

$$\sigma = F_\sigma(\sum_n \mathbf{z_n}/N), \qquad \mathbf{c_0}, \psi, \mathbf{h} = F_{\mathbf{c_0}}(\sum_n (\mathbf{z}_n; \gamma(d))/N), \qquad (1)$$

where $F_\sigma$ and $F_{\mathbf{c_0}}$ are MLPs consisting of four linear layers respectively, and $\gamma : \mathbb{R}^{3 \to 6 \times l}$ is a positional encoding of viewing direction $d \in \mathbb{R}^3$ as in Mildenhall et al. (2020) with $2 \times l$ different basis functions. $\mathbf{h}$ is the intermediate color feature extracted from the second to last layer of $F_{\mathbf{c_0}}$ that is later used in the appearance blending (section 3.2).

## 3.2 LEARNABLE APPEARANCE BLENDING

While our implicit body NeRF representation helps in robustness to geometric variation of arbitrary human shapes, it is difficult for such a general NeRF representation to convey all the high-frequency details (color and textures) present in the source view images. To remedy this, we leverage ideas from image-based rendering techniques Chen & Williams (1993); Buehler et al. (2001); Gortler et al. (1996); Levoy & Hanrahan (1996); Debevec et al. (1998); Hedman et al. (2018); Wang et al. (2021). In particular, a query point $\mathbf{x}$ can be projected to $N$ source views to directly retrieve the color values. We propose an appearance blending module $F_B$ that learns to predict blending weights among the $N$ source view colors $\mathbf{c}_n$ and the NeRF generated color $\mathbf{c}_0$.

The inputs to this network are the pixel-aligned features $p_n$ for $\mathbf{x}$, the relative viewing direction $\hat{d}_n$ which is defined by the difference and dot product between the target viewing direction $d$ and the source viewing direction $d_n$, the visibility $o_n$ of $\mathbf{x}$ with respect to each source view, the intermediate NeRF color feature $\mathbf{h}$ and the color confidence $\psi$ from $F_{c_0}$. We define the view index for the implicit body NeRF representation as $n{=}0$. Formally, the blending weights $w_n^{rgb}$ are computed as:

$$\{w_n^{rgb}\}_{n=0}^N = F_B(\{p_n; \hat{d}_n; o_n\}_{n=1}^N, \mathbf{h}, \psi). \qquad (2)$$

Note that our visibility $o_n$ is computed *without* the need for any ground truth 3D geometry. To pursue our scalable and practical setting without the reliance on the 3D supervision, we propose to exploit the underlying parametric body model to approximate the visibility for each query point. Specifically, we borrow the vertex visibility of the SMPL model. As in Huang et al. (2020); Bhatnagar et al. (2020); Peng et al. (2021a), we first search the SMPL mesh surface that is closest to the query point. Then, we define the visibility as a float value $o_n \in [0, 1]$ computed by the barycentric interpolation of the 0-or-1 visibility values of three vertices on the corresponding mesh facet. This is different from Cheng et al. (2022), where visibility trained with groundtruth is required. Also, our design differs from Zhao et al. (2022), where the groundtruth-supervised depth is utilized to sample source view colors. The final color $\mathbf{c}$ for $\mathbf{x}$ is computed via softmax as $\mathbf{c} = \sum_{n=0}^N (\exp(w_n^{rgb})\mathbf{c}_n)/(\sum_{i=0}^N \exp(w_i^{rgb}))$.

## 3.3 RE-POSING NIA FOR POSE ANIMATION

We describe how our proposed NIA can be extended to animate an unseen subject given new target poses at inference without any further training. Different from the previous works Chen et al. (2021); Peng et al. (2021a; 2022), where a canonical feature space for appearance retrieval is optimized in a per-subject manner, our generalization approach aims at reposing the NIA avatars that are created on the fly given a set of sparse reference images of unseen human subjects. Since the available source images in the reference space cannot be directly used in the target pose space (*i.e.*, observation space), the construction of our NIA representations should be considered with the deformation between the target and reference spaces.

To build the implicit body feature volume in the target pose space, we retrieve the relevant pixel-aligned features from the location of the reference SMPL vertices. By using the known SMPL correspondences, these pixel-aligned feature points are relocated and attached to the corresponding target pose SMPL vertices. The following sparse convolutions will result in the feature volume in the target pose space, where each body feature $z_n^{tar}$ is sampled from. The density and color in the target space are computed similarly to Eq. (1), except we use $z_n^{tar}$ that is transformed into the reference space, instead of $z_n$.

The appearance blending is defined similarly to Eq. (2) using the target-to-reference transformed query location $\mathbf{x}'$ and viewing direction $d'$, instead of $\mathbf{x}$ and $d$. The visibility is computed with respect to $\mathbf{x}'$ and the reference SMPL body.

**Deformation modeling.** We model the deformation $T$ from the target pose space (observation space) to reference space as the skeleton-driven deformation, which is the composition of the inverse and forward linear-blend skinning Lewis et al. (2000). Specifically, we can transform the query $\mathbf{x}$ in the target space into the reference space location $\mathbf{x}'$ through $T(\mathbf{x}) = (T^{fwd} \circ T^{inv})(\mathbf{x})$, where $T^{inv}$ and $T^{fwd}$ are defined as:

$$T^{inv} = \left( \sum_{k=1}^{K} w_k^{tar}(\mathbf{x}) G_k \right)^{-1} \mathbf{x} = \mathbf{x}^{can}, \qquad T^{fwd} = \left( \sum_{k=1}^{K} w_k^{can}(\mathbf{x}^{can}) G_k \right)^{-1} \mathbf{x}^{can} = \mathbf{x}', \quad (3)$$

where $\{G_k\} \in SE(3)$ is the $K$ transformation matrices produced by SMPL skeleton with $K$ body parts, and $w_k^{tar}$ and $w_k^{can}$ are the blending weights for $k$-th part sampled in the target and canonical space, respectively. Similar to the visibility computation (section 3.2), the blending weight is interpolated from the blending weights of the nearest SMPL vertices.

The deformation $\tilde{T}$ for the viewing direction, which transforms the target-space viewing direction $\mathbf{d}$ to the reference space viewing direction $\mathbf{d}'$ is defined in a similar manner. We denote the weighted sum of transformation matrices of $T$ as $[R(\mathbf{x}); t(\mathbf{x})]$. Then, $\tilde{T}(\mathbf{d})$ is defined as $\tilde{T}(\mathbf{d}) = R(\mathbf{x})\mathbf{d} = \mathbf{d}'$,

### 3.4 Volume Rendering and Loss Function

**Volume rendering.** To decide the color of each ray, we accumulate the density and color predictions along the ray $\mathbf{r}(t) = \mathbf{r}_0 + t\mathbf{d}$ for $t \in [t_{\text{near}}, t_{\text{far}}]$ as defined in NeRF Mildenhall et al. (2020) as follows:

$$\mathbf{C}(r) = \int_{t_{\text{near}}}^{t_{\text{far}}} \mathbf{T}(t)\sigma(\mathbf{r}(t))\mathbf{c}(\mathbf{r}(t), \mathbf{d})dt, \quad where \quad \mathbf{T}(t) = \exp\left( -\int_{t_{\text{near}}}^{t} \sigma(\mathbf{r}(s))ds \right) \quad (4)$$

In practice, we uniformly sample a set of 64 points $t \sim [t_{near}, t_{far}]$. The bounds for ray sampling $t_{near}, t_{far}$ are drived by computing the 3D bounding box of the SMPL body.

**Loss function.** Together with the final blended appearance $\mathbf{c}$, we also supervise the NeRF regressed $\mathbf{c_0}$ with a simple photometric loss to improve the prediction quality: $\mathcal{L} = \|\mathbf{c}_0 - \hat{\mathbf{c}}\|_2 + \|\mathbf{c} - \hat{\mathbf{c}}\|_2$ where $\hat{\mathbf{c}}$ denotes the groundtruth pixel color.

## 4 Experimental Results

To demonstrate the effectiveness of our proposed NIA method, we conduct experiments for novel view synthesis and pose animation tasks. We use ZJU-MoCap Peng et al. (2021b) for both tasks and ablation studies. Then we study our cross-dataset generalization ability by training on ZJU-Mocap and testing on MonoCap datasets without any finetuning.

### 4.1 Identity Generalization

**Competing baselines.** For novel view synthesis, the state-of-the-art competitors are NHP Kwon et al. (2021) and GP-NeRF Chen et al. (2022) which robustly perform generalization with the body surface feature. Other generalizable methods are pixel-NeRF Yu et al. (2020), PVA Raj et al. (2021b), Image-Based Rendering Network (IBRNet) Wang et al. (2021), and Keypoint NeRF Mihajlovic et al. (2022). We also compare with per-subject optimization method Neural Body Peng et al. (2021b), Neural Textures (NT) Thies et al. (2019), and Neural Human Rendering (NHR) Wu et al. (2020).

For pose animation task, we compare with the per-subject animatable NeRF methods which are Neural Body Peng et al. (2021b), A-NeRF Su et al. (2021) and Animatable-NeRF Peng et al. (2021a).

Note that the comparison with per-subject methods in both tasks is for reference purpose and places our (and other) generalizable method in disadvantage. These methods require one model (network) trained for a single subject, *i.e.*, as many models as the number of testing subjects, and they are

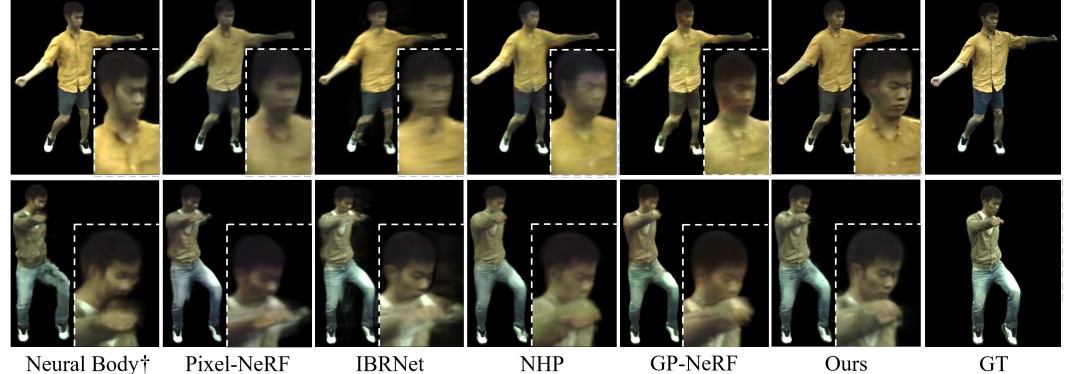

Neural Body†        Pixel-NeRF        IBRNet        NHP        GP-NeRF        Ours        GT

Figure 2: Novel view synthesis results on ZJU-MoCap dataset. †: Neural Body Peng et al. (2021b) is a per-subject optimized method, and is tested on the seen subjects with unseen poses. Pixel-NeRF Yu et al. (2020), IBRNet Wang et al. (2021), NHP Kwon et al. (2021) and GP-NeRF Chen et al. (2022) are generalizable methods which are tested on unseen subjects with unseen poses.

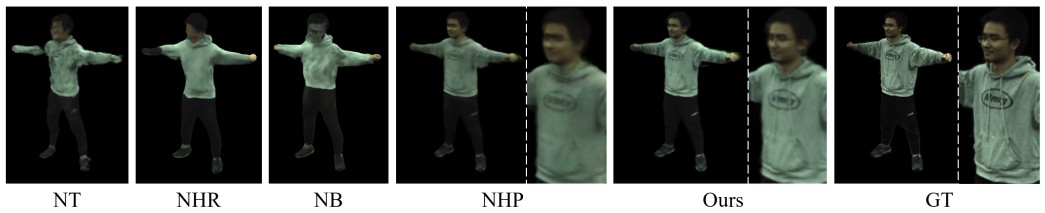

NT              NHR              NB              NHP              Ours              GT

Figure 3: Results on the unseen poses of seen subjects on ZJU-MoCap dataset. NT: Neural Texture Thies et al. (2019), NHR: Multi-view Neural Human Rendering Wu et al. (2020), NB: Neural Body Peng et al. (2021b), NHP: Neural Human Performer Kwon et al. (2021), and NIA (ours). NT, NHR, NB are per-subject optimized methods.

| method | PSNR | SSIM |
|---|---|---|
| Training: **per-subject** | | |
| NT | 22.28 | 0.8720 |
| NHR | 22.31 | 0.8710 |
| NB | 23.79 | 0.8870 |
| Training: **generalizable** | | |
| NHP | 26.94 | 0.9290 |
| **NIA (ours)** | **27.57** | **0.9398** |

a. Eval. on **seen** subjects, unseen poses.

| method | PSNR | SSIM |
|---|---|---|
| PVA | 23.15 | 0.8663 |
| Pixel-NeRF | 23.17 | 0.8693 |
| IBRNet | 24.54 | 0.8935 |
| NHP | 24.75 | 0.9058 |
| GP-NeRF | 24.49 | 0.9012 |
| Keypoint NeRF | 25.03 | 0.8969 |
| **NIA (ours)** | **25.79** | **0.9200** |

b. Training: **generalizable** (*one model for multiple subjects*); Eval. on **unseen** subjects, unseen poses.

Table 1: Novel view synthesis on ZJU-MoCap. All are evaluated on the same testing sequences.

evaluated on the training seen subjects with unseen poses. On the other hand, the generalizable methods train only one model which is evaluated on all unseen subjects with unseen poses.

### 4.1.1 NOVEL VIEW SYNTHESIS

**Setup.** The ZJU-MoCap dataset contains 10 human subjects with 23 synchronized cameras. We follow the same training and testing protocols as in Kwon et al. (2021).

**Results.** The quantitative comparison of our NIA and other methods is shown in Table. 1. Our model achieves state-of-the-art performance on ZJU-Mocap dataset. Especially, the previous state-of-the-art NHP Kwon et al. (2021) relies on temporally aggregated features to compensate the sparse input views. Our NIA outperforms NHP by +1.0 dB PSNR and +1.5% SSIM, while utilizing only

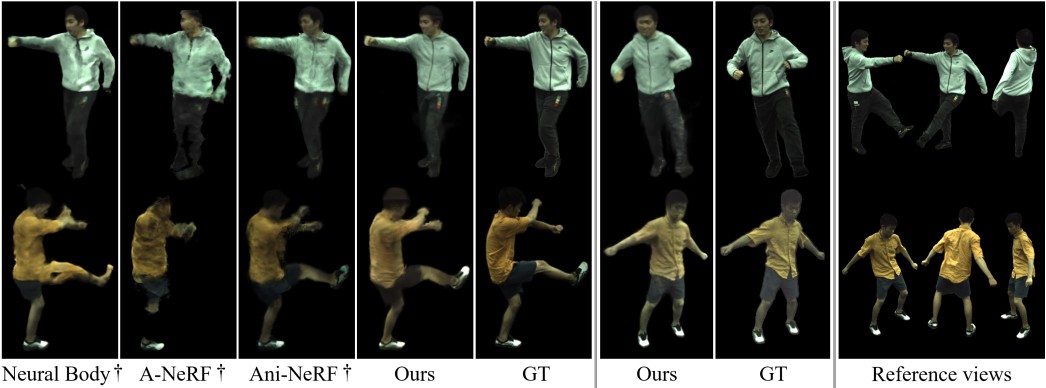

| Neural Body † | A-NeRF † | Ani-NeRF † | Ours | GT | Ours | GT | Reference views |

Figure 4: Pose animation results on ZJU-MoCap dataset. †: Neural Body Peng et al. (2021b), A-NeRF Su et al. (2021), and Ani-NeRF Peng et al. (2021a) are per-subject optimized methods, and are evaluated in novel pose synthesis for the seen subjects. Our NIA is the only generalizable method, and is evaluated in novel pose synthesis given only three-view reference images of unseen subjects. Column 6 and 7 are our results for another poses, and the ground truths. The last columns are the reference image input to our NIA.

still images. NIA also outperforms the most recent state-of-the-art methods GP-NeRF Chen et al. (2022) and Keypoint NeRF Mihajlovic et al. (2022) by +1.3 / +0.7 dB PSNR and +2.0 / +2.5% SSIM.

In Table. 1 and figure 3, we show the performance on the unseen poses of the seen subjects. Despite of the disadvantageous generalizable setting (one network for all subjects), our method significantly outperforms other baselines including the per-subject optimized methods.

As shown in figure 2, our NIA exhibits more high-frequency details compared to NHP and GP-NeRF. Other generalizable methods that utilize image-conditioned features (pixel-NeRF, PVA, and IBRNet) often suffer noisy surface prediction, while NIA shows more robust results on different human shapes.

### 4.1.2 POSE ANIMATION

**Setup.** We evaluate our NIA for pose animation task on ZJU-MoCap dataset. Given a set of sparse view reference images of an unseen person, our on-the-fly generated avatar for that person is reposed to new target poses. We also use three input views for this task.

While we do perform generalization onto unseen subject identities, we compare with per-subject methods. Note that this comparison puts our method on *disadvantage* and thus is only to provide a reference level, since the competing methods have *seen* the testing subjects (no identity generalization for baselines).

Specifically, we compare with Neural Body Peng et al. (2021b), A-NeRF Su et al. (2021) and Animatable-NeRF Peng et al. (2021a) which made their code public and reported their results on ZJU-MoCap dataset. Meanwhile, we acknowledge there are more recent and advanced per-subject animation methods Su et al. (2022); Zheng et al. (2022); Peng et al. (2022); Weng et al. (2022). For all methods, we evaluate on the same testing sequences as in the novel view synthesis task.

**Results.** Table. 2 shows the quantitative comparison. Remarkably, our NIA model that is generalized onto unseen subjects outperforms the per-subject optimized competitors tested on seen subjects by healthy margins: +1.2 dB / +2.0% over Neural Body, +0.1 dB / +1.2% over A-NeRF and +0.1 dB / +1.1% over Animatable-NeRF in PSNR and SSIM, respectively. In figure 4, we observe that both Neural Body and Animatable-NeRF produce noisy and blurry texture output on novel poses, even though their models are trained to memorize the appearance of the same human subject. Our NIA method preserves relatively more high-frequency details like clothes wrinkles and textures. Note that it is a very challenging setting that requires our model to instantly create animatable avatars on the fly from only three reference view images of unseen human subjects.

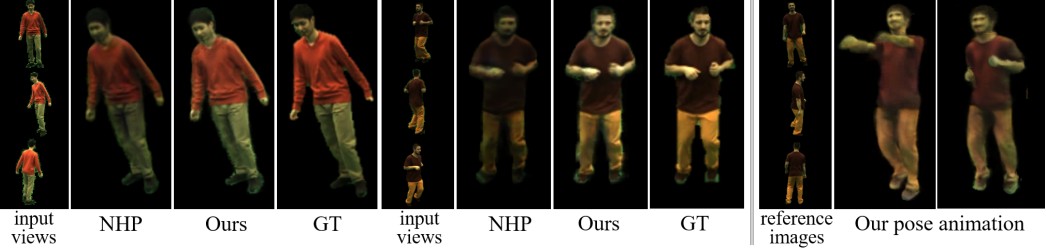

input views | NHP | Ours | GT | input views | NHP | Ours | GT | reference images | Our pose animation

Figure 5: Cross-dataset generalization results without finetuning. Our NIA outperforms NHP Kwon et al. (2021) on novel view synthesis, when trained on ZJU-MoCap and tested on MonoCap datasets (left). Our NIA can also f perform pose animation of unseen subjects given only a sparse set of still images (right).

| method | training | evaluation | PSNR | SSIM |
|---|---|---|---|---|
| Neural Body | | | 22.88 | 0.8800 |
| A-NeRF | per-subject | **seen** subjects, unseen poses | 23.91 | 0.8893 |
| Animatable-NeRF | | | 23.88 | 0.8901 |
| **NIA (ours)** | generalizable | **unseen** subjects, unseen poses | **24.04** | **0.9000** |

Table 2: Pose animation on ZJU-MoCap dataset. All methods are evaluated on the same testing sequences and target poses. A-NeRF Su et al. (2021), Animatable-NeRF Peng et al. (2021a), and Ours.

| method | PSNR | SSIM |
|---|---|---|
| NHP | 17.51 | 0.7457 |
| **NIA (ours)** | **19.12** | **0.7804** |

Table 3: **Cross-dataset** generalization (train on ZJU-MoCap, and test on MonoCap.

## 4.2 CROSS-DATASET GENERALIZATION

For the out-of-domain, cross-dataset generalization, we train a model on the ZJU-MoCap dataset, and test on the MonoCap dataset without any finetuning. MonoCap consists of DeepCap Habermann et al. (2020) and DynaCap Habermann et al. (2021) dataset sequences that are captured by dense camera views. Given the different lighting and capture settings between the ZJU-MoCap and MonoCap datasets (*e.g.*, dark v. bright studio), our NIA improves the cross-dataset generalization over NHP by clear margins of +1.6 dB PSNR and +4.5 % SSIM in Table. 3.

As shown in the visual results (figure 5), the implicit body NeRF prediction of NHP especially suffers from the different color distribution between these datasets.

On the other hand, our NIA is a hybrid model combining the implicit body NeRF with the direct pixel fetching from the input images (*i.e.*image-based rendering), which helps to generalize to out-of-domain MonoCap datasets. Given only three snaps of a new person, our NIA is able to perform plausible pose animation as shown in figure 5 (right: 'pose animation').

## 4.3 ABLATION STUDIES AND ANALYSIS

We provide ablation studies on ZJU-MoCap dataset by evaluating novel view synthesis performance of our NIA variants on unseen subject identities with unseen poses.

**NIA architecture designs.** We study the contribution of different modules of our NIA network in Table. 4.i and figure 6. First, we ablate the two main components of our method: implicit body representation (*a*) and image-based rendering (*b*). By removing each component, we observe a performance drop of -1.63 dB / -3.80% (*a*) and -1.55 dB / -1.49% (*b*) in PSNR and SSIM respectively. figure 6 visualizes how each component contributes to the rendering quality. Without the implicit body representation, the model tends to suffer inaccurate shape prediction especially on the regions fully occluded from all source views (*e.g.*, ghosting artifact near the right hand). On the other hand, removing the image-based rendering leads to blurry results and the loss of high-frequency texture details. Our full NIA model-(*e*) leverages these complementary merits and shows the best results both quantitatively and qualitatively.

We further investigate the impact of our neural appearance blending module. It learns to blend predictions from the implicit body representation (*i.e.*NeRF-predicted color) and the color values

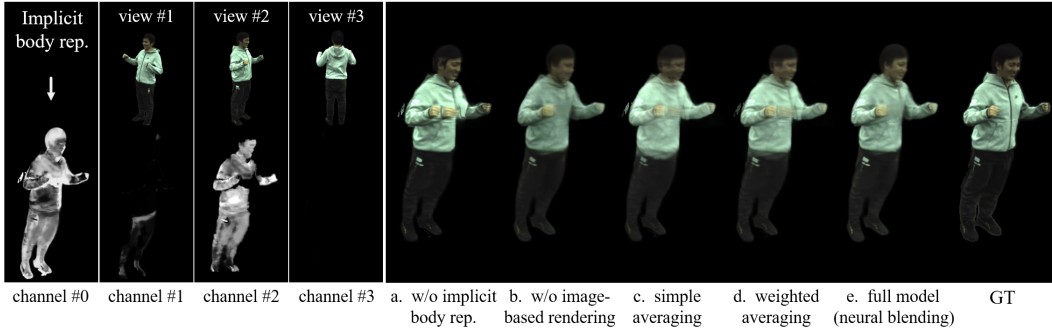

channel #0    channel #1    channel #2    channel #3    a. w/o implicit    b. w/o image-    c. simple    d. weighted    e. full model    GT
body rep.    based rendering    averaging    averaging    (neural blending)

Figure 6: Input views (#1,2,3) and the corresponding channels of learned neural blending weight maps. Channel #0 corresponds to the prediction from the implicit body representation (NeRF). Remaining columns show the results of ablation items in Table. 4.i and the ground truth.

| NIA variant | PSNR | SSIM |
|---|---|---|
| a.    w/o implicit body representation | 24.62 | 0.8888 |
| b.    w/o image-based rendering | 24.70 | 0.9119 |
| c.    w/o neural appearance blending (simple averaging) | 25.28 | 0.9168 |
| d.    w/o neural appearance blending (weighted averaging) | 25.70 | 0.9236 |
| e.    full model | **26.25** | **0.9268** |

i. NIA network components.

| # views | PSNR | SSIM |
|---|---|---|
| 1 | 24.71 | 0.9006 |
| 2 | 25.53 | 0.9150 |
| 3 | 26.25 | 0.9268 |

ii. number of input views.

Table 4: Ablation studies on ZJU-MoCap dataset; novel view synthesis task.

from the source images (*i.e.*, image-based rendering), adaptively to different regions to render. Instead of our learnable design, we experiment by replacing it with two plausible deterministic functions. The first is a naive simple averaging of different color predictions (*c*), and the second is a weighted averaging based on the cosine similarity between each source view's viewing direction and the target view (*d*). The comparison in rows-(*c, d, e*) verifies the benefit of our learnable appearance blender, demonstrated by the gains of +0.97 dB / +1.0% and +0.45 dB / +0.3% in PSNR and SSIM over the two deterministic variants respectively. figure 6 also shows the contribution of adaptive blending that helps resolving the blurs and ghosting artifacts.

**Number of camera views.** Table. 4.ii shows that the performance of our NIA gradually increases with more source views. Increasing the number of source views can provide our model with more relevant observations from the views closer to the target view.

**Visualizations.** In figure 6, we visualize the learned blending weight maps for all source views (#1, 2, and 3) and the color prediction from the learned implicit body representation (denoted as channel #0). Among the source views, we observe that the view #2 which is the most similar to the target view, is highly weighted in large area. Meanwhile, the implicit body representation takes charge of the overall content regions while especially highlighting the occluded regions that are not visible from all source views, (*e.g.*, the left boundary of the torso, and behind the right hand). With our neural appearance blending, our NIA is able to properly blend the available observations and the learned NeRF prediction.

## 5    CONCLUSION

We introduce Neural Image-based Avatars (NIA), a novel approach for feed-forward avatar creation from sparse view images of unseen persons. Our key idea is to learn to adaptively combine the parametric body model-based NeRF and image-based rendering techniques. Our rendering results in novel view synthesis and pose animation tasks exhibit improved robustness and fine details on different human subjects and poses. Notably, our NIA outperforms the state-of-the-art competitors, and even per-subject optimized methods in both tasks. We believe our approach is a step forward in generalizing the creation of human avatars capable of their free-view synthesis and pose animation.

## 6 ACKNOWLEDGMENTS

We thank Sida Peng of Zhejiang University for many very helpful discussions on a variety of implementation details of the Animatable-NeRF. We thank Shih-Yang Su of University of British Columbia for helpful discussions on the A-NeRF details. We thank Prof. Helge Rhodin of UBC and his group for the insightful discussions on the human performance capture. This work was partially supported by National Science Foundation Award 2107454.

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
