# OpenReview forum: "Neural Image-based Avatars: Generalizable Radiance Fields for Human Avatar Modeling"
_ICLR.cc/2023/Conference — ICLR 2023 poster_

### Official Review · Reviewer_Ke1f · 2022-10-23

**Confidence:** 3
**Correctness:** 4
**Technical Novelty And Significance:** 3
**Empirical Novelty And Significance:** 2
**Recommendation:** 8

**Clarity, Quality, Novelty And Reproducibility:**

The paper is clear and technically sound. The overall pipeline is new, but it strongly relies on Kwon et al. (2021) and Wang et al. (2021). The authors describe the implementation details in the supplementary material and claim that the code will be published upon publication.

Minor suggestions:

- section(s) -> Section(s); sec. -> Sec..
- Section 4.1.1 could be on top of page 6.
- Eval. -> Evaluation/Evaluated (Tab.1).
- References could be included in Table 1, or at least in its caption.

**Strength And Weaknesses:**

Main Strengths:
- The method gathers two complementary state-of-the-art approaches (i.e. neural radiance fields and neural image-based rendering).
- The NIA is generalizable for different subjects.

Main Weaknesses:
- The approach relies on sparse view images and the authors do not refer to the LOLNeRF (Rebain et al., CVPR22), a related and recent method based on single views.
- Most of the previous approaches rely only on PSNR and SSIM metrics. Despite that, it would be interesting to see additional metrics on the evaluation, or at least to read the authors' comments on why other metrics were not adopted.



**Summary Of The Paper:**

The authors propose Neural Image-based Avatars (NIA), a method for synthesizing novel views and poses. Its main advantages are the implicit body NeRF representation and the image-based rendering. They claim to outperform the current state-of-the-art on the ZJU-MoCap and MonoCap datasets.

**Summary Of The Review:**

The paper introduces a new approach to generalized novel views and poses synthesis. Although it mainly put together existing methods (Kwon et al. (2021) and Wang et al. (2021)), it achieves state-of-the-art results. Therefore, I believe the contribution is worthy of publication.

---

> ### Author Response · Authors · 2022-11-18
> **Reply to Reviewer Ke1f**
>
> We appreciate the reviewer for the thoughtful feedback and are happy that the reviewer acknowledge the contribution of our method. Please see our visual results in neural_image_based_avatars_gen-Supplementary Material.zip-ID_5716_supple-rebuttal.pdf.
> * * *
> **1. Citation**
>
> Thank you for the suggestion. We will cite and discuss LOLNeRF in the paper.
>
> * * *
> **2. Metric**
>
> As the reviewer acknowledged, we followed the evaluation practice of other human NeRF works [1,2,3,4,5,6]. Following the suggestion, we additionally report the perception-based metric LPIPS [7] on the generalization task compared with the main competitor NHP[3]. We outperform NHP by 10.5% LPIPS (lower the better). Thank you for the constructive suggestion, and we will include this score in our final revision.
>
> | Method     | PSNR  | SSIM   | LPIPS |
> |------------|-------|--------|-------|
> | NHP        | 24.75 | 0.9058 | 0.142 |
> | NIA (ours) | **26.25** | **0.9268** | **0.127** |
>
> * * *
> **3. Minor suggestions**
>
> Thank you for the helpful and constructive feedback. We will modify the final version accordingly.
>
> [1] Peng et al. “Neural Body” CVPR (2021).
>
> [2] Peng et al. “Animatable NeRF” ICCV (2021).
>
> [3] Kwon et al. “Neural human performer” NeurIPS (2021).
>
> [4] Chen et al. “GP-NeRF” ECCV (2022).
>
> [5] Su et al. “A-NeRF” NeurIPS (2021).
>
> [6] Su et al. “DANBO” ECCV (2022).
>
> [7] Zhang et al. “ The unreasonable effectiveness of deep features as a perceptual metric” CVPR (2018).

---

### Official Review · Reviewer_Pdgp · 2022-10-24

**Confidence:** 4
**Correctness:** 3
**Technical Novelty And Significance:** 2
**Empirical Novelty And Significance:** 2
**Recommendation:** 3

**Clarity, Quality, Novelty And Reproducibility:**

Clarity is not sufficient.

The framework of NIA is similar to [Zhao et al. (2022)], weakening its novelty.

Reproducibility is limited due to its complex system.

**Strength And Weaknesses:**

[Strengths]
+ The motivation is well-written and easy to follow.
+ The citations are sufficient.
+ The authors provide several visualizations for realizing their rendering results.

[Weaknesses]
- The proposed idea novelty is limited. The framework of NIA is similar to [Zhao et al. (2022)] since the two major steps of both methods are generalizable implicit body representation and sparse views blending. The core design of NIA is the neural appearance blending module (sec. 3.2), yet no ablation study to discuss and analyze why such a design helps.
- Three assumptions exist in this work: 1) the calibration parameters of the multi-view input images are known; 2) the foreground human region masks are known; 3) the fitted SMPL models are available as prior. The first two assumptions are also seen in other methods; however, yet the third assumption is new in this work. The third assumption seems too strong since the SMPL model estimation is a vital factor affecting human NeRF rendering. Therefore, while assuming the fitted SMPL models are available as a prior, it is doubtful that the comparison experiments are fair enough compared with other methods. Notice that Table 4 (i) shows that the NIA w/o image-based rendering (with the third assumption) has already better than most other methods.
- In sec 3.3, the description of “the 3D coordinates of the posed SMPL vertices in both reference and target spaces are known by the nature of motion-tracked SMPL model” is another strong assumption that makes the NIA available to avoid the finetuning step. It is unclear why the such assumption is available while dealing with the unseen subject with unseen poses.
- The comparison experiments are not convincing. It seems that some existing methods are reimplemented since the reported results of these methods (degraded) in this paper are not the same as their published ones. It is better to annotate which results are reimplemented and which are not.

- Some implementation details are not clear:
1) The limitations of the sparse input images. Are there any constraints for view directions?
2) The feature representation of P_n is not defined;
3) The design of two MLPs in equation (1) is different from NeRF [Mildenhall et al. (2020)]. Why is such a design reasonable, and what is its advantages?
4) The architecture of F_B in equation (2). Does (2) means that it has one single model for a specific value of N?
5) SE(3), x^can, k-th part sampling method, function R(), function t() in sec. 3.3;

Reference:
[Zhao et al. (2022)] Fuqiang Zhao, Wei Yang, Jiakai Zhang, Pei Lin, Yingliang Zhang, Jingyi Yu, and Lan Xu. Humannerf: Efficiently generated human radiance field from sparse inputs. In CVPR, 2022.


**Summary Of The Paper:**

This paper aims to obtain Neural Image-based Avatars (NIA) for synthesizing novel views and poses of an unseen human from sparse input images. NIA is derived from a neural blending network to integrate 1) NeRF-based body representation conditioned on parametric SMPL human body model and 2) neural image-based rendering.

**Summary Of The Review:**

The primary concern of this paper is its limited novelty. The key idea is to learn to adaptively combine the parametric body model-based NeRF and image-based rendering techniques, which largely overlap the method proposed by Zhao et al. (2022). Besides, the assumptions of "the fitted SMPL models are available as prior" and "the 3D coordinates of the posed SMPL vertices in both reference and target spaces are known" are too strong to compare with the existing methods for the fairness concern.

---

> ### Author Response · Authors · 2022-11-18
> **Reply to Reviewer Pdgp (1/4)**
>
> We appreciate the reviewer for the feedback. Please see our visual results in neural_image_based_avatars_gen-Supplementary Material.zip-ID_5716_supple-rebuttal.pdf. Also, due to the space limit, we will continue our replies in the following comments.
> * * *
> **1. Novelty**
>
> Comparison to Zhao et al. [1]: Zhao et al. requires 3D depth supervision for learning the neural blending. Also, it requires per-subject finetuning to deal with new identities and new poses. Also, we use a different implicit body representation for better synthesis.
>
> * Depth supervision - a fundamental difference in assumption about the geometry prior.
>
> One of the fundamental differences of Zhao et al. is the use of 3D depth groundtruth as supervision. In contrast, our method uses the estimated SMPL (ie, from EasyMocap[2], same as in  [3,4,5,6]), which is a weaker prior yet more practical assumption than having the 3D depth supervision. This leads to different challenges, and accordingly the following design choices and generalizability between these two methods:
>
> * Choice of implicit body representation in the blending module
>
> Zhao et al. uses a pixel-aligned features-conditioned NeRF as their implicit body representation. However, without the 3D supervision, the pixel-aligned features alone are prone to errors under complex poses as reported in Zhao et al., NHP[5], also as observed in our experiment (below table and rebuttal.pdf-Figure 2).
>
> To remedy this, we leverage recent advances in SMPL-based NeRF methods [3,5,6], and adopt the skeletal body representation (namely, skeletal feature NeRF) [5,6] as our implicit body representation. Specifically, the skeletal body representation is obtained by anchoring the pixel aligned features to the SMPL vertices, then diffusing the anchored features to the 3D space (via SparseConv[7]) to sample at the query location. We show the choice of the implicit body representation (i.e., pixel-aligned feature NeRF vs skeletal feature NeRF) is critical in the blending module and the final results ~ (below table and rebuttal.pdf-Figure 2)
>
> We will clarify this in our revision.
>
> | Implicit body representation | PSNR  | SSIM   |
> |------------------------------|-------|--------|
> | Pixel-aligned features-based NeRF   | 25.92 | 0.9172 |
> | Skeletal features-based NeRF        | **26.25** | **0.9268** |
>
> * Visibility computation
>
> Another difference lies in the way of computing the visibility of each query location (with respect to the input views), which is input to the appearance blending module. While Zhao et al. relies on the 3D depth supervision to know the visibility information, we propose a 3D supervision-free method and take the visibilities of the nearest SMPL vertice to the query location as an approximation. The visibility of the SMPL vertice is obtained by rasterization. We demonstrate our blending module can be trained reasonably well with this approximated method.
>
> * Generalization ability - per-subject finetuning vs zero-shot generalization
>
> Despite using stronger 3D supervision at training, Zhao et al. requires per-subject finetuning at test time. Their novel-view synthesis “results without finetuning are low-detailed and blurry” as stated in their paper [1]. Especially for pose animation, their learnt deformation field is not generalizable and also needs the per-subject finetuning. In contrast, we show that blending the raw RGB pixel values and the color prediction from the SMPL-based skeletal features is generalizable in both novel view synthesis and pose animation. Please note that we use the common per-frame estimated SMPL as in [3,4] without special motion tracking techniques, yet we observe the generalization in pose animation.  Our trained NIA model is able to deal with unseen subjects on-the-fly without any finetuning. Last but not least, we also demonstrated NIA’s performance in a more challenging cross-dataset generalization-both in novel view and novel pose synthesis setting, which is under-explored setting in the field.

---

> > ### Author Response · Authors · 2022-11-19
> > **Reply to Reviewer Pdgp (2/4)**
> >
> > * * *
> > **2. Ablation study on neural appearance blending module**
> >
> > We ablate the inputs to our neural blending model in the below table. Note that in the below table-(1,2,3) and Table.6-(c,d) in the main paper blend NeRF-color and pixel colors. However, (1,2,3) use learnable appearance blending to predict the weight between NeRF and pixel color, while (c,d) uses deterministic blending.
> >
> > We found that adding the NeRF intermediate feature and visibility together to the inputs performs the best (3), so we adopt it as our final inputs. The relative viewing direction (difference and dot product between the target and input viewing direction), can provide information on how relevant each input view is to the target view. Also, the visibility indicates which input views are reliable. We additionally observed that adding the NeRF intermediate feature to the blending module input is beneficial.
> >
> > In addition, as we ablated in Table 4i in the main paper, we present that the learnable blending between the two rendering methods is more beneficial than deterministic blending schemes (simple/weighted sum according to the view distance, Table 4i-c,d and Figure 6-c,d), and showed qualitatively that it successfully blends the two rendering methods in  (Figure 6 - weight map visual).
> >
> > | Input variant                | PSNR  | SSIM   |
> > |------------------------------|-------|--------|
> > | 1. p + d                     | 25.82 | 0.9235 |
> > | 2. p + d + h                 | 26.01 | 0.9247 |
> > | 3. p + d + h + o (ours) | **26.25** | **0.9268** |
> >
> > p: pixel-aligned feature / d: relative viewing direction / h: NeRF intermediate feature / o: visibility
> >
> >
> > ***
> > **3. SMPL assumption. The comparison is not fair.**
> >
> > We summarize the different assumptions used in all compared methods in the table below. Please note that we use the same assumption that the SMPL parameter for each frame is available, following the standard of recent approaches [3,4,8,5,6].
> > As also stated in our supplementary material, we use the same SMPL fit provided by the ZJU-MoCap [3] dataset and use EasyMoCap[2] to obtain the SMPL parameter for the DeepCap[9]/DynaCap[10] dataset. EasyMoCap[2] is the same tool the author of the ZJU-MoCap[3] dataset used: https://github.com/zju3dv/EasyMocap.
> >
> >
> > Regarding the comparison, we use NT[11] and NHR[12] that are reimplemented by Neural Body paper. These versions compute the mesh (NT) and point clouds (NHR) using the SMPL vertices. Note that other recent methods  [4,14,15] also compare with the reimplemented version of NT and NHR.
> >
> >
> > In the table below, our method outperforms the best existing SMPL-based NeRF methods (and all other methods), which suggests our gain is not solely from the use of SMPL fit.
> >
> > * Train: per-subject (one model per subject) / Eval: seen subjects, unseen poses (Table 1-a in the main paper)
> >
> > | Method | Assumption          | PSNR  | SSIM   |
> > |--------|---------------------|-------|--------|
> > | NT     | Mesh (SMPL)         | 22.28 | 0.8720 |
> > | NHR    | Point clouds (SMPL) | 22.31 | 0.8710 |
> > | NB     | SMPL                | 22.88 | 0.8800 |
> > | NHP    | SMPL                | 26.46 | 0.9270 |
> > | NIA (ours)    | SMPL                | **27.85** | **0.9500** |
> >
> > * Train: generalizable (one model for multiple subjects) / Eval: unseen subjects, unseen poses (Table 1-b in the main paper)
> >
> > | Method       | Assumption | PSNR      | SSIM       |
> > |--------------|------------|-----------|------------|
> > | PVA          | -          | 23.15     | 0.8663     |
> > | Pixel-NeRF   | -          | 23.17     | 0.8693     |
> > | IBRNet       | -          | 24.54     | 0.8935     |
> > | KeypointNeRF | Keypoints  | 25.03     | 0.8969     |
> > | NHP          | SMPL       | 24.75     | 0.9058     |
> > | GP-NeRF      | SMPL       | 24.49     | 0.9012     |
> > | NIA (ours)         | SMPL       | **26.25** | **0.9268** |
> >
> > ***
> > **4. “The 3D coordinates of the posed SMPL vertices in both reference and target spaces are known by the nature of motion-tracked SMPL model” is another strong assumption.**
> >
> > To clarify, we did not use any additional motion-tracking techniques. We used the per-frame estimated SMPL parameters (as discussed above).
> >
> > What we meant by “motion-tracked SMPL model” is as follows (we will revise the explanation).
> > We followed the common assumption made in other animatable NeRF works [3,4,8] and generalizable NeRF works [1,5,6] that the SMPL parameter for any frame (including reference and target spaces) is available. The SMPL vertice are automatically computed from the SMPL parameters  as in [3,5,6].

---

> > > ### Author Response · Authors · 2022-11-19
> > > **Reply to Reviewer Pdgp (3/4)**
> > >
> > > * * *
> > > **5.Comparison is not convincing**
> > >
> > > In the table below, we summarize the reference of the numbers in our comparison Table 1. We note that we use the other methods' official code with the provided weight checkpoint, or take the numbers from the NHP paper.
> > >
> > > As mentioned in supplementary material, we used the same train/test split used in the NHP paper. However, the GP-NeRF [6] paper and its official code and checkpoint is based on a different train/test split. Therefore, we train the GP-NeRF with the splits used in NHP (and all other methods) using their official code without any other modifications, and report the numbers. The subject IDs of the used train/test split (ZJU-Mocap [3] dataset) are as follows: train (313,315,377,386,390,392,396) / test (387, 393, 394).
> > >
> > > | Method       | Reference                       | PSNR      | SSIM       |
> > > |--------------|---------------------------------|-----------|------------|
> > > | PVA          | from NHP paper Table 2                | 23.15     | 0.8663     |
> > > | Pixel-NeRF   | from NHP paper Table 2          | 23.17     | 0.8693     |
> > > | IBRNet       | Used their official code        | 24.54     | 0.8935     |
> > > | KeypointNeRF | from KeypointNeRF paper Table 4 | 25.03     | 0.8969     |
> > > | NHP          | from NHP paper Table 2          | 24.75     | 0.9058     |
> > > | GP-NeRF      | Used their official code        | 24.49     | 0.9012     |
> > > | NIA (ours)   | -                               | **26.25** | **0.9268** |
> > >
> > > * * *
> > > **6. Implementation details**
> > >
> > > * Are there any constraints for viewing direction?
> > >
> > > There are no constraints on the viewing direction.  However, as the target viewpoint deviates from the input viewpoint, the synthesis quality degrades and gets blurry. An extreme example with a single input view  is shown in rebuttal.pdf-Figure 1.
> > >
> > > * The feature representation of P_n is not defined
> > >
> > > As written in the second sentence of section 3.1 in the main paper, it is defined as follows: Given N available source views, we extract pixel-aligned image features from each view n (n = 1 ~ N) and attach them to the SMPL vertices to construct a per-view coarse feature set Pn.
> > >
> > > * The design of two MLPs in equation (1) is different from NeRF [17]. Why is such a design reasonable, and what is its advantages?
> > >
> > > For the input to the density and color MLP, we use the same input (ie skeletal body feature) as NHP. The advantage of skeletal body features over the query 3D coordinates (as in Mildenhall et al.[17]) is that it provides human prior and thus leads to robust reconstruction given sparse input views (see NHP paper for more details).  For the output, our color MLP additionally outputs the NeRF intermediate feature h (see Section 3.1 last paragraph) that is later fed into our neural blending module. We show in the first table of the second reply (Reply to Reviewer Pdgp(2/4))  that feeding the NeRF intermediate feature h leads to the performance gain.
> > >
> > > * The architecture of F_B in equation (2). Does (2) means that it has one single model for a specific value of N?
> > >
> > > F_B is a single trained network that can handle different values of N. For the number of input views less than 3, we repeat the closest view to the target view. For example, when N=1, the input is repeated three times. When N=2, we repeat the closer one to the target view twice. When N>3, we choose the closest three views. We include all mentioned cases in our training.
> > >
> > > * SE(3), x^can, k-th part sampling method, function R(), function t() in sec. 3.3;
> > >
> > > -SE(3) is a rigid transformation represented with a transformation matrix consisting of a translation and rotation in 3-D.
> > >
> > > -x_can is the query point in the target space transformed into the canonical space.
> > >
> > > -Blending weight for k-th part sampling method: We used the same method as in Animatable NeRF [4], where we first find the closest surface point on the SMPL mesh. Then the blend weight is computed by performing the barycentric interpolation of the blend weights of three vertices on the corresponding mesh facet. For the more details, please refer to section 3.2 of [4].
> > >
> > > -R is the rotation matrix of T (T is the deformation from the target pose space to reference space), t is the translation vector of T.
> > >
> > > * * *
> > > **7. Reproducibility**
> > >
> > > As stated in the supplementary material, we will make the code public upon publication. All the used datasets are already publicly available: ZJU-MoCap [3], DynaCap [10], DeepCap[9].

---

> > > > ### Author Response · Authors · 2022-11-19
> > > > **Reply to Reviewer Pdgp (4/4)**
> > > >
> > > > * * *
> > > >
> > > > [1] Zhao et al. “HumanNeRF” CVPR (2022).
> > > >
> > > > [2] Easy Mocap. “https://github.com/zju3dv/EasyMocap”
> > > >
> > > > [3] Peng et al. “Neural Body” CVPR (2021).
> > > >
> > > > [4] Peng et al. “Animatable NeRF” ICCV (2021).
> > > >
> > > > [5] Kwon et al. “Neural human performer” NeurIPS (2021).
> > > >
> > > > [6] Chen et al. “GP-NeRF” ECCV (2022).
> > > >
> > > > [7] Liu et al. “Sparse convolutional neural networks” CVPR (2015).
> > > >
> > > > [8] Liu et al. “Neural Actor” SIG ASIA (2021).
> > > >
> > > > [9] Habermann et al. “DeepCap” CVPR (2020).
> > > >
> > > > [10] Habermann et al. “DynaCap” TOG (2021).
> > > >
> > > > [11] Thies et al. "Deferred neural rendering" TOG (2019).
> > > >
> > > > [12] Wu et al. “Multi-view neural human rendering” CVPR (2020).
> > > >
> > > > [14] Su et al. “A-NeRF” NeurIPS (2021).
> > > >
> > > > [15] Su et al. “DANBO” ECCV (2022).
> > > >
> > > > [16] Raj et al. “PVA” CVPR (2021).
> > > >
> > > > [17] Mildenhall et al. “NeRF” ECCV (2020).

---

> > > > > ### Comment · Reviewer_Pdgp · 2022-11-19
> > > > > **Main concerns are not entirely addressed.**
> > > > >
> > > > > The reviewer thanks the authors for their responses. However, the main concerns are not entirely addressed in their responses.
> > > > > 1. The novelty of the proposed idea is limited since the claimed key ingredient, as shown in the abstract and introduction, of NIA is a hybrid appearance blending module that combines the advantages of the implicit body NeRF representation and image-based rendering. The hybrid framework of NIA is similar to [Zhao et al. (2022)] though NIA provides some helpful implementation replacements to make such a hybrid framework stronger.
> > > > > 2. The comparison experiments are not convincing since the reported results in Table 1 (a) are different from that in the Neural Body paper (“Regarding the comparison, we use NT[11] and NHR[12] that are re-implemented by Neural Body paper.”) Especially, the results in [Table 1, Zhao et al. (2022)] show the better performance.

---

> > > > > > ### Author Response · Authors · 2022-11-19
> > > > > > **Different numbers due to different settings (seen vs unseen pose evaluation, 6 vs 3 input views)**
> > > > > >
> > > > > >  Thank you for the review, and let us respond to Q2 and Q1.
> > > > > > ***
> > > > > > **Reply to Q2 : Different numbers due to different settings (seen vs unseen pose evaluation, 6 vs 3 input views)**
> > > > > >
> > > > > >  NeuralBody [3] and Zhao et al [1]. evaluated in a different setting to ours.
> > > > > > The numbers in Neural Body Table 2 and Zhao et al. Table 1 are evaluated on the **seen pose**.
> > > > > > Our number in the Table 1-a is evaluated on the **unseen pose**.
> > > > > >
> > > > > > While the evaluation setting is already different (seen vs unseen poses), another notable difference is that Zhao et al. uses **6** input views while we use **3** input views.
> > > > > >
> > > > > > The response to Q1 is added in the following comment.

---

> > > > > > > ### Author Response · Authors · 2022-11-20
> > > > > > > **Reply to Q1**
> > > > > > >
> > > > > > > Thank you for the feedback.
> > > > > > > ***
> > > > > > > **1. Clarification**
> > > > > > >
> > > > > > > We will revise the abstract as follows:
> > > > > > >
> > > > > > > "A key ingredient of our method is a hybrid appearance blending module that combines the advantages of the ~~implicit body NeRF representation~~ **skeletal body NeRF representation** and image-based rendering."
> > > > > > >
> > > > > > > which aligns with the following sentence in our introduction (sec 1-third paragraph):
> > > > > > >
> > > > > > > “ It is a hybrid framework that combines body surface feature conditioned NeRF (e.g., Kwon et al. (2021)) and image-based rendering techniques (e.g., Wang et al. (2021)).”
> > > > > > >
> > > > > > > ***
> > > > > > > **2. We will revise and add more discussion on Zhao et al.**
> > > > > > >
> > > > > > > We will strengthen the acknowledgement and discussion about the related work [Zhao et al.]. We will also discuss and clarify the differences between the two methods as described in the rebuttal (e.g., 1. 3D depth GT (Zhao et al.) vs SMPL estimate (ours), 2. implicit body representation: pixel feature (Zhao et al.) vs skeletal feature (ours), 3. visibility computation, 4. per-subject finetuning (Zhao et al.) vs generalizable-and-animatable (ours), 5. six-view (Zhao et al.) vs three-view (ours)).
> > > > > > >
> > > > > > > ***
> > > > > > > **3. Importance of our replacement/modification in generalization and sparse-view setting**
> > > > > > >
> > > > > > > It is worth noting that the above-mentioned replacements (1 - 4) not only improve the overall quality, but play a critical role in enabling our generalizable-and-animatable human rendering, unlike Zhao et al. which requires per-subject finetuning and 3D depth supervision. This is also ablated in the rebuttal.
> > > > > > >
> > > > > > > The importance of our modification is also demonstrated by the number of input views used in the two methods. Zhao et al. uses 6 input views while we use sparse 3 views. In particular, Zhao et al. shows steep degradation as the input views reduces from 6 to 4 views (-6.7 PSNR) or to 2 views (-10.1 PSNR) (see Zhao et al. supp. Table 4). Furthermore, Zhao et al. suffers geometry errors with 2 input views (Fig.9 of Zhao et al.), while we preserve the geometry even with a single input view (Figure 1 in our rebuttal).
> > > > > > >
> > > > > > >
> > > > > > > ***
> > > > > > > **4.Our method is generalizable-and-animatable without per-subject finetuning and 3D supervision, unlike Zhao et al.**
> > > > > > >
> > > > > > > Last but not least, please note that we tackle a generalizable-and-animatable setting without any per-subject finetuning. It is an extremely challenging and under-explored setting, which in turn requires different design choices than those using the per-subject finetunining (Zhao et al). We demonstrate the benefit of the proposed method with unseen subjects, unseen poses and pose animation. We further exhibit our generalization-and-animation ability onto out-of-domain data, i.e., cross-dataset setting. We hope our contributions could be considered based on our advances in the generalization and animation performance.

---

### Official Review · Reviewer_28Gy · 2022-10-24

**Confidence:** 4
**Correctness:** 3
**Technical Novelty And Significance:** 3
**Empirical Novelty And Significance:** 3
**Recommendation:** 6

**Clarity, Quality, Novelty And Reproducibility:**

* **Clarity**: Most contents are clear and easy to follow.
* **Quality**: The proposed method has been evaluated and studied on substantial benchmarks and settings. Especially, experiments are also performed on cross-dataset settings.
* **Novelty**: The method directly combines two rendering methods, which both exist in previous works. This makes the novelty a little bit in discount.
* **Reproducibility**: Most implementation details are provided. It is likely to reproduce the method. Code is not available.

**Strength And Weaknesses:**

**Strength**
* This method can achieve good results on unseen views and unseen poses. Thanks to the image-based rendering method, it has good generalization performance across datasets.
* The ablation experiments show that the fusion of implicit body representations and image-based rendering plays a complementary role.
* Sufficient experiments have proved the effectiveness of the method.


**Weaknesses**
* When the number of viewing angles is reduced to 1, the results are greatly degraded. Which part of this degradation comes from deserves further analysis, e.g. with regard to multi-view consistency.
* The appearance blending module shows effectiveness in visualization and quantitative studies. However, why it works or the mechanism it works is not demonstrated clearly.
* The method directly combines two rendering methods, which both exist in previous works. Any new insight about the pipeline/module design needs to be clarified.

**Summary Of The Paper:**

This paper combines two rendering modules for synthesizing novel views and novel poses of human bodies from sparse multi-view images.
The first one is a body representation based on neural radiance fields and SMPL, extracting pixel-to-image features from each view, and attaching features to SMPL vertices. In this way, a coarse feature map for each view can be constructed, which is further converted into dense feature voxels through convolutions. Multi-view features are obtained through view-wise self-attention processing, and colors are finally obtained through MLPs.
The second one is a neural image-based rendering module, which can improve the details of the result by directly obtaining the pixel color from the source view image.
Results of the two modules are finally combined together with the prediction weights.

**Summary Of The Review:**

Through extensive experiments, this paper shows better performance than previous methods. And made a preliminary exploration on cross-dataset settings. Some questions/issues need to be fixed. The influence of the view number and the mechanism of appearance blending need to be clarified. Any new insight about the pipeline/module design needs to be further discussed to rebut the novelty concern.

---

> ### Author Response · Authors · 2022-11-18
> **Reply to Reviewer 28Gy**
>
> We appreciate the reviewer for the thoughtful and constructive feedback. Please see our visual results in neural_image_based_avatars_gen-Supplementary Material.zip-ID_5716_supple-rebuttal.pdf.
> * * *
> **1. Performance on the single input view**
>
> We present the novel view synthesis result given a single input view in rebuttal.pdf-Figure 1. We observe that the synthesis results on the adjacent views to the input view show reasonable multi-view consistency. However, as the target view deviates from the input view (e.g., opposite to the input view), the synthesis quality degrades and becomes blurry. This in turn leads to the degradation in the numerical results, when taking average over all views. The reason why is that our implicit body representation is mainly trained to interpolate between the given sparse views, rather than to hallucinate the unseen contents from scratch.
>
> * * *
> **2. Discussion on the appearance blending module**
>
> We ablate the inputs to our neural blending model in the table below. The relative viewing direction (difference and dot product between the target and input viewing direction), can provide information on how relevant each input view is to the target view. Also, the visibility indicates which input views are reliable. We additionally observed that adding the NeRF intermediate feature to the blending module input is beneficial.
>
> | Input variant                | PSNR  | SSIM   |
> |------------------------------|-------|--------|
> | 1. p + d                     | 25.82 | 0.9235 |
> | 2. p + d + h                 | 26.01 | 0.9247 |
> | 3. p + d + h + o (ours) | **26.25** | **0.9268** |
>
> p: pixel-aligned feature / d: relative viewing direction / h: NeRF intermediate feature / o: visibility
>
> * * *
> **3. Insight about the pipeline/module design**
>
> Although our method combines the existing rendering methods, namely, the implicit body representation and image-based rendering, it was non-trivial to choose what implicit body representation to combine with. One possible option as the implicit body representation is the pixel-aligned features-based NeRF. However, we observed combining the pixel-aligned features-based NeRF prediction leads to unsatisfactory results under complex poses (See rebuttal.pdf-Figure 2 and below table). Therefore, we adopt SMPL-based skeletal feature representation (e.g., NHP[1]) as the implicit body representation, and show it can successfully complement the image-based rendering given difficult poses and very sparse views.
>
> In addition, as we ablated in Table 4i, we present that the learnable blending between the two rendering methods is more beneficial than deterministic blending schemes (simple/weighted sum according to the view distance, Table 4i-c,d and Figure 6-c,d), and showed qualitatively that it successfully blends the two rendering methods in  (Figure 6 - weight map visual).
> Last but not least, we show that our learnt combination of the skeletal feature representation and the image-based rendering can perform reasonably in the ‘generalizable-and-animatable’ setting and cross-dataset generalization-and-animation without any per-subject finetuning, which is under-explored setting in the field.
>
> | Implicit body representation | PSNR  | SSIM   |
> |------------------------------|-------|--------|
> | Pixel-aligned features-based NeRF   | 25.92 | 0.9172 |
> | Skeletal features-based NeRF        | **26.25** | **0.9268** |
>
> * * *
> **4. Reproducibility**
>
> The code will be made public upon publication as stated in the supplementary material.
>
> [1] Kwon et al. “Neural human performer” NeurIPS (2021).

---

### Official Review · Reviewer_4ShL · 2022-10-25

**Confidence:** 4
**Correctness:** 4
**Technical Novelty And Significance:** 3
**Empirical Novelty And Significance:** 4
**Recommendation:** 8

**Clarity, Quality, Novelty And Reproducibility:**

The paper is clearly-written and shows high quality. I believe the work is original as far as I'm aware.

**Strength And Weaknesses:**

Paper strengths:
- The proposed method is novel in combining human NeRF and image-based rendering.
- The proposed method shows clearly superior performance both qualitatively and quantitatively.
- Experimental evaluations are extensive and comprehensive.

Paper weaknesses:
- It seems unclear how much the current model's performance depends on accurate SMPL fitting. Would the proposed model be compatible with in-the-wild selfie videos followed by a single-view SMPL estimation?
- I am a bit confused by the feature pooling in Equation (1) which currently does not consider visibility, should invisible views be ignored here?

**Summary Of The Paper:**

The paper presents a human full-body avatar model that combines human NeRF and image-based rendering methods. The proposed model achieves significant improvement for unseen pose and unseen person synthesis. Extensive quantitative and qualitative evaluations have been conducted to measure the model's effectiveness.

**Summary Of The Review:**

Overall speaking I think this is a solid paper. The method is novel, the results are significantly better than previous state-of-the-art, and the evaluations are thorough. My concerns are minor and can be addressed by additional discussions. Therefore, my initial rating for the paper is accept.

---

> ### Author Response · Authors · 2022-11-18
> **Reply to Reviewer 4ShL**
>
> We appreciate the reviewer for the thoughtful feedback and are happy that the reviewer acknowledge the novelty of our method. Please see our visual results in neural_image_based_avatars_gen-Supplementary Material.zip-ID_5716_supple-rebuttal.pdf.
> * * *
> **1. Dependency on accurate SMPL fitting**
>
> As we stated in the Section 4 - Limitations (supp.), the dependency on the accurate SMPL fit is one of our limitations. If the SMPL fit deviates much from the original geometry (in terms of pose or shape), the same SMPL vertice will be projected to non-corresponding pixels across different views/time. This will cause the blending of misaligned pixel colors, which in turn generates blurry results. In fact, this limitation is shared by all other recent SMPL-based NeRF works [2,3,4,5]. We agree it is an interesting research question how to deal with low-quality SMPL fit in the generalization setting (e.g., jointly optimize SMPL fit refinement as in Chen et al.[1]). In our paper, we focus on the generalizable-and-animatable setting, an under-explored topic, using the SMPL fit commonly used in the recent related works.
>
> Also, we present the novel view synthesis result given the single input view in rebuttal.pdf-Figure 1. We can see that the synthesis results on the views that are adjacent to the input view are reasonable. However, as the target viewpoint deviates from the input viewpoint, the synthesis quality degrades. Note that we use the SMPL provided by the ZJU-Mocap dataset, which is computed by multi-view estimation.
>
>
> * * *
> **2. Clarification on the feature pooling**
>
> Thank you for giving us the opportunity to clarify. Given the SMPL fits and the camera parameters we compute the visible vertices by rasterization, as stated in the section 2.1- Input details (supp.).  Then we use image features only for these visible vertices when constructing the body feature P_n (in section 3.1). z_n is built from P_n via self-attention between visible features, which was computed already considering the visibility. z_n is then divided by the number of valid views. We will clarify this in the revision.
>
> [1] Chen et al. "Animatable Neural Radiance Fields from Monocular RGB Videos" arXiv (2021).
>
> [2] Peng et al. “Neural Body” CVPR (2021).
>
> [3] Peng et al. “Animatable NeRF” ICCV (2021).
>
> [4] Kwon et al. “Neural human performer” NeurIPS (2021).
>
> [5] Chen et al. “GP-NeRF” ECCV (2022).

---

> > ### Comment · Reviewer_4ShL · 2022-11-21
> > **Response to rebuttal**
> >
> > I appreciate the authors' extended effort to improve the paper and clarify potential issues. My main concerns have been successfully addressed. Due to the dependency on very accurate SMPL tracking, I think the method is not yet entirely usable in practice yet. This said I do see the paper making meaningful research progress. Therefore, my final rating is 7.

---

> > > ### Author Response · Authors · 2022-12-01
> > > **Reply to Reviewer 4ShL: Inaccurate SMPL + single input view result**
> > >
> > > We appreciate the reviewer for acknowledging the contribution of our paper. To clarify, the SMPL tracking does not necessarily have to be very accurate. Our method can handle the reasonable amount of SMPL error (both pose and shape errors) unless it deviates too much from the original geometry. We are attaching the anonymous link for the visualization: https://anonymous.4open.science/r/re-587C/iclr_23_rebuttal_figure.png
> > >  As can be seen in the figure, our method performs well given only the single view input with inaccurate SMPL (the SMPL error is shown by the input image overlaid with SMPL fit). We will also include the visualization in the final revision.

---

### Decision · Program_Chairs · 2023-01-20

**Decision:**

Accept: poster

**Justification For Why Not Higher Score:**

The overall pipeline is relatively incremental, which strongly relies on Kwon et al. (2021) and Wang et al. (2021).

**Justification For Why Not Lower Score:**

The proposed method has achieved compelling results on generalized novel view and pose synthesis.

**Metareview: Summary, Strengths And Weaknesses:**

This paper was reviewed by three experts in the field. Based on the reviewers' feedback, the decision is to recommend the paper for acceptance to ICLR 2023. All the reviewers acknowledged the quality achieved by the proposed method for generalized novel view and pose synthesis. The reviewers did raise some valuable concerns that should be addressed in the final camera-ready version of the paper, e.g., adding more discussions and ablations to similar works such as Zhao et al. (2022). The authors are encouraged to make the necessary changes to the best of their ability. We congratulate the authors on the acceptance of their paper!


**Note From Pc:**

if the above contains the word "oral" or "spotlight" please see: "oral" presentation means -> notable-top-5% and "spotlight" means -> notable-top-25%. As stated in our emails, we are disassociating presentation type from AC recommendations

**Summary Of Ac-Reviewer Meeting:**

N/A